# Biomimetic Coatings Obtained by Combinatorial Laser Technologies

**Emanuel Axente [1]**, **Livia Elena Sima [2]** and **Felix Sima [1],***

[1] Center for Advanced Laser Technologies (CETAL), National Institute for Laser, Plasma and Radiation Physics (INFLPR), 409 Atomistilor, Magurele 077125, Romania; emanuel.axente@inflpr.ro

[2] Department of Molecular Cell Biology, Institute of Biochemistry of the Romanian Academy, 296 Splaiul Independentei, Bucharest 060031, Romania; lsima@biochim.ro

\* Correspondence: felix.sima@inflpr.ro; Tel.: +4-021-457-4243

**Abstract:** The modification of implant devices with biocompatible coatings has become necessary as a consequence of premature loosening of prosthesis. This is caused mainly by chronic inflammation or allergies that are triggered by implant wear, production of abrasion particles, and/or release of metallic ions from the implantable device surface. Specific to the implant tissue destination, it could require coatings with specific features in order to provide optimal osseointegration. Pulsed laser deposition (PLD) became a well-known physical vapor deposition technology that has been successfully applied to a large variety of biocompatible inorganic coatings for biomedical prosthetic applications. Matrix assisted pulsed laser evaporation (MAPLE) is a PLD-derived technology used for depositions of thin organic material coatings. In an attempt to surpass solvent related difficulties, when different solvents are used for blending various organic materials, combinatorial MAPLE was proposed to grow thin hybrid coatings, assembled in a gradient of composition. We review herein the evolution of the laser technological process and capabilities of growing thin bio-coatings with emphasis on blended or multilayered biomimetic combinations. These can be used either as implant surfaces with enhanced bioactivity for accelerating orthopedic integration and tissue regeneration or combinatorial bio-platforms for cancer research.

**Keywords:** bio-coatings; biomimetics; laser deposition; PLD; MAPLE; tissue engineering; cancer

## 1. Biomimetic Materials

During the last two decades, new challenges in nanoscience and nanotechnology have been continuously addressed, in particular within the biomedical field [1–3]. The necessity of new biomaterials with improved properties have led to interdisciplinary studies at the interface between physics, chemistry, materials science, biology, and medicine [1]. There are, by now, four generations of biomaterials with the latter one capable of adapting to extra- and intra-cellular processes that could allow for the understanding of signaling pathways mediating inter- and intra-cellular communications [4–6]. However, there is still a breach in the in-depth understanding of nanomaterial interactions with biological entities both in vitro and in vivo. The design of innovative biomaterials should aim to precisely control the composition–properties relationship in order to modulate cell behavior in the field of tissue engineering and regenerative nanomedicine [7] or cancer theranostics [8].

With the development of the materials science field, biomaterial surface properties have advanced from bio-inert to bio-active and bio-resorbable, then to bio-conductive. These characteristics have been shown to strongly influence cell behaviors such as viability, proliferation, migration, and differentiation [4,5,9]. Biomimetic materials are, in addition, considered to effectively control cell behavior through partial replication of specific tissue features [10]. In a simple description, biomimetics

rely on the ability to create synthetic characteristics from nature-inspired shapes and principles with the aim to achieve the desired biological responses. As a consequence, an improvement in the biomaterial surface could be attained by modifying their basic morphological properties to create a biomimetic environment that could eventually control the cell–surface interaction. In addition, in order to structurally and functionally recapitulate the natural micro-environment, inorganic–organic composite structures become necessary. Hence, a broad variety of coating materials are extensively explored for tissue engineering applications, spanning from ceramics, natural and synthetic biopolymers, proteins, peptides, enzymes, and growth factors to cover *composite biomimetic materials* consisting of blends with drugs or other biomolecules.

## 2. Biomimetic Coatings: In Vitro Testing Strategies and Clinical Trials

This section focuses on tissue engineering applications of bio-coatings, although few examples will highlight cancer research. The main materials used in implantology are metals, ceramics, composites, and polymers. In the case of prosthetic and dental applications, the uses of bio-coatings are anticipated to improve osseointegration of metal implants, while preventing allergies and chronic inflammation. The main culprit for these clinical manifestations is the corrosion of metals and alloys used in clinics for orthopedic, orthodontic, cardiovascular, or craniofacial metallic implant fabrication (mainly titanium-based alloys). The main techniques used to enhance the corrosion resistance of biomaterials were reviewed by Asri et al. [11].

First attempts at covering the surface of stainless-steel implants with hydroxyapatite (HA, $Ca_{10}(PO_4)_6(OH)_2$) by electrochemical deposition were contradictory and somehow disappointing in terms of resistance to corrosion [12,13]. This bioactive material has, however, the advantage of resemblance to the inorganic chemical structure of bone and teeth. Therefore, numerous studies have been dedicated to enhance biomaterial properties by coating inert metallic substrates with HA, ion-substituted calcium phosphates (CaPs), or hybrid organic-inorganic coatings containing CaPs [14].

### 2.1. In Vitro Testing of Inorganic Coatings for Tissue Engineering

CaPs are not osteoinductive (able to induce *de novo* heterotopic bone formation) *per se* unless they are structured in porous structures [15]. It has been suggested that such topography induces osteogenesis by regulating primary cellular cilia length and transforming growth factor (TGF) receptor recruitment [16]. Recently, molecular cell analyses have revealed that plasma cell glycoprotein 1 (PC-1) is a key protein responsible for the osteoinductive response of cells to CaP ceramics as a negative regulator of bone morphogenic protein-2 (BMP-2) signaling, a key pathway governing bone development and regeneration [17]. The proposed molecular driver of osteoinductivity is the localized depletion of calcium and phosphate ions from body fluid [18].

It is now accepted that composition, stiffness, and topography are key biophysical features that modulate bone progenitor cell differentiation [19]. Different processing techniques have been tested for their capacity to induce the proper bone formation response in osteoblasts and mesenchymal stem cells (MSCs) when exposed to CaP-based ceramic coatings and composites thereof.

A comparison between plasma electrolytically oxidized (PEO) and plasma-sprayed HA coatings on a Ti-6Al-4V alloy revealed that the PEO obtained films induced a significant increase in collagen production by human MG-63 osteoblast-like cells, while they had 78% lower surface roughness compared to plasma spraying [20]. Recent research has helped in successfully overcoming the low adhesive bond strength of the plasma sprayed HA coating over metallic substrates by creating an interfacial layer consisting of a gradient HA coating prepared by laser engineered net shaping (LENS™) followed by plasma spray deposition [21].

Researchers have next tested the possibility of substituting the calcium ions in HA or other CaP by biologically relevant ions or compounds. These are known to provide support for both osteogenesis and angiogenesis processes [22]. Studies led by Adriana Bigi et al. proposed and tested in vitro several doping solutions for CaP-based laser deposited coatings containing either strontium (Sr) alone [23]

or in combination with magnesium (Mg) [24] or with the drug, zoledronate (ZOL) [25]. For this work, different approaches were employed for coating synthesis and further subjected to in vitro testing with osteoblasts and osteoclasts. These studies have shown an improvement in osteoblast adhesion, proliferation, and differentiation, while osteoclast proliferation decreased with an increase in Sr content of the Sr-HA PLD coatings [23]. Mg and Sr doping of octacalcium phosphate (OCP) enhanced osteoblast proliferation, activity, and differentiation [24]. Testing of reciprocal gradients of Sr and ZOL doped HA in osteoblast-osteoclast co-cultures has revealed associative conditions that favor bone production activity more than each of the dopants alone [25]. Experiments showed that ZOL promotes collagen type I (COLL1) production, whereas Sr significantly increases the production of alkaline phosphatase (ALP), two osteoblast markers expressed during late and early differentiation, respectively. Sr was more effective in enhancing osteoblast viability and activity, while ZOL was more effective in inhibiting osteoclastogenesis. The laser method allowed, in this case, the modulation of the coatings' impact on bone cell activity by "titrating" the gradient proportion in which the two molecules produce most beneficial outcome, with minimal cytotoxicity.

Inductively coupled radio frequency (RF) plasma spray was also used to generate Sr–HA and Mg–HA coatings on Ti. Their performance was investigated using human fetal osteoblasts and results showed an increased cell attachment and proliferation as well as improved differentiation in the presence of Sr when compared to HA and Mg-HA [26].

In vivo experiments in a rat model of osteoporosis have confirmed the improved implant osseointegration when Ti was coated by Sr-doped HA when compared to Zn and Mg dopants [27]. Sr–HA bonding with animal bone was almost twice as strong as for HA alone, as proven by the biomechanical tests. This result was supported by the increased new bone formation surrounding the implant, when HA contained ion dopants. The coating was obtained, however, through electrochemical deposition; there are no similar in vivo reports on laser obtained Sr–HA coatings. Similar improvements in bone regeneration were reported for 5.7% Mg–HA when tested on New Zealand White rabbits using the granulate as filling for a femoral bone defect compared to HA alone [28].

Corrosion is not the only problem related to implant success. Another pathophysiologic event that prevents long-term stability of the implanted biomaterial is the loosening of the prosthesis due to its encapsulation by a fibrotic scarring tissue. Therefore, besides the coatings used to prevent metallic corrosion, solutions have been developed to increase the osseointegration of implants, especially those addressing long bones replacement, where forces applied on the connecting joints are higher and increase bonding strength to compensate them is much needed. Mechanical stimulation is impeded at the implantation site, which negatively impacts new bone formation by the surrounding tissue. This blockage in mechanotrasduction, due to decreased mechanical loading, is known as "stress shielding" and is reduced by the use of flexible materials in the fabrication of implant stems [29].

A representative study for this approach is the one proposed by Scarisoreanu et al. [30]. In this work, the laser synthesis of a functional biocompatible piezoelectric material coating on a flexible Kapton polymer substrate is reported. The experiments validate such structures as optimal support for MSC adhesion, proliferation, and osteogenic differentiation, which is in favor of potential future use of piezoceramic coatings on bone implants.

Efforts to predict the fibrotic tendency of a biomaterial through a comparative multi-parametric in vitro model have revealed a high correlation between an in vitro test based on autologous plasma to in vivo-obtained biomaterial assessments [31]. It is foreseen that a combination of a three-dimensional fibrin matrix and primary macrophage assays would help identify promising biomaterials and decrease the need for animal studies.

## 2.2. In Vitro Testing of Organic Coatings for Tissue Engineering

Osteoprogenitor cells response to their microenvironment is largely mediated by signals received from the extracellular matrix (ECM) (reviewed in [32]) that are transmitted to the cell nucleus *via* actionable signaling pathways. Hence, stem cell fate is modulated epigenetically upon integration of

signals from outside the cell. Recent findings indicated that this mechanism can be coerced into diverting stem cell commitment to desired differentiation pathways: biochemical pre-treatment of MSCs with either of the five identified epigenetic modulators (Gemcitabine, Decitabine, I-CBP112, Chidamide, and SIRT1/2 inhibitor IV) enhanced osteogenesis in vitro [33]. Out of these, Gemcitabine and Chidamide successfully rescued the loss in osteogenic differentiation potential in MSCs obtained from aged donors, hence providing indication of the potential use of these small molecules as pharmacological agents against bone demineralization.

Similar effects are searched for when screening different biomaterials for prosthetics/bone regeneration purposes. In this case, the interaction of MSCs with the biomaterial surface is mediated by mechanotransduction pathways that engage cell integrin receptors binding to ECM molecules adsorbed on the substrate. The ideal implant biomaterial would provide the necessary cues in favor of osteogenic differentiation while discouraging cell commitment alternatives.

The *top–down* approach of generating scaffolds from tissues by decellularization and/or intermediary ECM purification steps largely preserves the complex composition of the in vivo niche, while the *bottom–up* synthetic approach uses key elements of the bone tissue able to induce osteogenic commitment in order to support the process of endochondral ossification involved in bone healing [34,35]. While the first strategy is fitted mainly for filling bone defects, the second is amenable to the development of biomimetic implant coatings. A purely biomaterial-based solution for the induction of endochondral ossification proposed for the challenging regeneration of critical-size defects was recently reported for the first time [36]. This involves the use of porcine collagen as a scaffold with channel-like pore architecture to assist the first steps of bone healing through extracellular matrix alignment, CD146$^+$ progenitor cell accumulation, and restrained vascularization. The recapitulation of developmental bone growth process was demonstrated in human stromal cell culture and in rat models of bone regeneration, where the macroporous scaffold functioned as a highly aligned biomaterial template that aligns ECM fibers along the bone axis. Alternatively, a synthetic ECM was also proposed as a hybrid between hyaluronan–fibrin and a polymer biodegradable template represented by Poly(Lactic-co-Glycolic Acid) (PLGA) microspheres [37].

In order to create coatings with increased bone tissue biomimetism, synthetic polymers or proteins are tested in conjugation or not with inorganic compounds. They are either biodegradable [38], and designed to provide temporary support for the regenerating tissue, or stable through the whole healing process [39]. Bioactive composite scaffolds combining nanofibrous polycaprolactone (PCL) matrix and HA or TCP as the mineral phase, either added inside the fiber structure or as a film obtained by electrospinning, represent a biomimetic material able to induce MSC osteogenesis, without the need of osteoinductive factor treatment [40]. Furthermore, carbonated HA–gelatin composite-coated PCL/TCP scaffolds were shown to stimulate osteogenesis compared to PCL/TCP alone [41]. ECM-mimetic scaffolds [42] and coatings [43] have been designed that contain growth factors or ECM proteins to be released in the healing environment upon biomaterial implantation in order to enhance tissue mineralization.

### 2.3. In Vitro Testing of Bio-Coatings for Cancer Research

Aside from the extensive research and development in bone tissue engineering field to implement the use of biomaterial coatings for specific needs, there are also a few recent examples of use in cancer research.

The development of biomimetic constructs by laser direct-write has allowed the proposal of a model to study breast cancer cell invasion into adipose tissue [44]. The construct reflects mammary microenvironment architecture through replication of the spatial relationship between cancer cells and tissue resident adipocytes, which are encapsulated in alginate–collagen microbeads. This tissue-like structure enables the investigation of the physiological contribution of obesity to breast cancer cell invasion. Another example represents a potential cell-based cancer immunotherapy application of biomaterials. For this application, magnetron sputtering was used to produce micropatterned nickel

titanium thin films loaded with ovarian cancer-specific CAR-T cells for local delivery into solid tumors to eradicate established multifocal disease [45].

It is rather accepted that none of the deposition techniques depicted above could fabricate an "*ideal coating*", able simultaneously to: (i) precisely mimic the native physiological micro-environment, and (ii) accurately control drugs or other functional molecule delivery from coatings to specific sites. Accordingly, the biomaterials research community has focused on the development and the fabrication of blended and multilayered bio-coatings in view of obtaining multiple functionalities, eventually required in clinical trials [46–49].

### 2.4. In Vivo Clinical Trials for Regenerative Medicine

Despite the multitude of research approaches to enhance orthopedic and dental implant osseointegration, the majority of those that reached the clinical trials stage are based on CaP coatings (recently reviewed in [50]). The main physical deposition techniques to produce CaP-based coatings are the thermal spray processes followed by other vaporization-based methods (PLD, MAPLE, ion-beam-assisted deposition (IBAD), RF magnetron sputtering) [51].

A search using the "bone implant coatings" keywords in the *ClinicalTrails.gov* database retrieved 71 entries. Out of these, 29 were completed clinical trials, while the others are still on-going. Three trials have been completed with results, out of which two tested plasma sprayed coated biomaterials. First is an HA plasma sprayed acetabular cup that was tested in parallel with a BoneMaster HA coated cup to compare the effect upon bone density and clinical outcomes in patients with total hip replacement (identifier NCT00859976). After two years post-operation, two patients in the plasma-sprayed shell receiving group had not fixated implants and another one showed an unstable implant out of 22 participants, while no radiolucency was revealed in the BoneMaster treated patients ($n = 12$). The other clinical assessments showed no significant differences between patient outcomes in the two groups. Second was a new Bone Anchored Hearing Aid (BAHA) device that was tested, consisting of a commercially pure titanium coated with a plasma sprayed HA layer on the entire soft tissue-contacting surface of the abutment up to 3 mm below the top surface. The randomized controlled trial took place in four European countries and was aimed at comparing this new bone conducting BA400 model implant with the traditional BA300 abutment (identifier NCT01796236, [52]). The use of the HA coated abutment allowed for soft tissue preservation and improved cosmetic results for patients as well as demonstrated cost savings. Third was a dental bone inductive implant obtained by adsorption of rhBMP2 on a porous titanium oxide surface (identifier NCT00422279, [53]). Four participants presenting alveolar ridge abnormality were enrolled in the study bearing two implants each. The aim was to evaluate implant stability and new bone formation around the implant. Based on preliminary experiments in canine models, a minimum dose of 15 and 30 μg rhBMP2 per implant was chosen. The assessment of implant stability at six months showed positive results for 4/4 implants in supra-alveolar position while 3/4 implants were stable upon implantation at extraction sites. However, no evidence of bone growth was seen in any of the cases, which rendered the study not successful.

Continuous efforts are still necessary for testing the various promising prosthetic materials in a clinical setting for the validation of their patient benefits.

## 3. Biomimetic Coatings: Advantages and Drawbacks of Processing Technologies

### 3.1. Biomimetic Processing Technologies

There are various physical and chemical deposition techniques employed for the synthesis of bio-coatings and they exhibit advantages and limitations regarding the type of material, the preservation of stoichiometry, control of morphological and structural properties, or processing of the coating area. Among them, plasma vapor deposition (PVD) techniques, usually apply to inorganic material coating. One may enumerate thermal evaporation, atomic layer deposition, electron beam evaporation,

sputter deposition in plasma, reactive sputter deposition, cathodic arc deposition, pulsed electron deposition, electroless plating, or laser deposition. On the other hand, there are also coating methods appropriate for organic materials such as the Langmuir–Blodgett dip coating, sol-gel, layer by layer deposition, aerosol spraying, dip coating, spin coating or laser evaporation that entail liquid solutions of the material in a volatile solvent. Typically, each method is suitable to a limited class of compounds, either inorganic or organic. This represents a limiting factor, particularly in the case of a composite bio-coating. Composites become necessary to solve specific medical problems such as to improve osseointegration or to decrease local tissue inflammation and avoid infection.

We reviewed several studies (Table 1) devoted to the deposition techniques for either inorganic, organic, or composite materials.

**Table 1.** The main deposition methods of inorganic, organic, and hybrid biocompatible coatings selected from the last years.

| Deposition Method | Advantages | Drawbacks | Bio-Coating Materials/Application |
|---|---|---|---|
| **Laser Techniques** | | | |
| Pulsed laser deposition (PLD) | Stoichiometric and adherent coatings, morphology control, easy to obtain multilayered thin films, good versatility of experimental design, thickness control. | Limited to inorganic coatings, small area covering (few cm), high costs. Micrometer-sized droplets and particulates on surfaces. | Inorganic coatings: HA [54,55], and Bioglass (BG) [56]/Implant devices. |
| Combinatorial-Pulsed laser deposition (C-PLD) | Preserve the properties of PLD, synthesis of combinatorial libraries of thin films, controlled doping of coatings, cover larger substrates (glass slide). | Limited to inorganic coatings, high costs. | Inorganic coatings: Ag-doped HA [57], Ag-doped Carbon [58]/Model surfaces. |
| Matrix assisted pulsed laser evaporation (MAPLE) | Applied to both organic and inorganic coatings, multilayers and multistructures, nanoparticulate films, thickness control. | Generation of micrometer-sized droplets and particulates on surfaces, small covering areas. | Inorganic, organic, hybrid coatings: HA [55], HA/Lactoferrin/polyethylene glycol-polycaprolactone copolymer [59], hybrid BG-biopolymer [60]/Implant devices, drug delivery. |
| Combinatorial-Matrix assisted pulsed laser evaporation (C-MAPLE) | Preserve the properties of MAPLE, single-step synthesis of combinatorial bio-coatings, suitable for organic, inorganic and composites, cover larger substrates (glass slide). | Generation of micrometer-sized droplets and particulates on surfaces, high costs. | Inorganic, organic, hybrid coatings: Chitosan/ bio-mimetic apatite [61], CaPs, biopolymers [62,63] Levan and Chitosan blends [64], Graphene Oxide, BSA protein, drugs [65]/Implant devices, drug delivery, model surfaces. |
| Laser-induced forward transfer (LIFT) | Micro-patterns with high spatial resolution, size and separation distance between structures, easy control. | Limited to patterns, difficulties for large area micro-fabrication, difficult to control the height of the patterns. | Inorganic, organic, hybrid micropatterns: collagen and nanoHA [66], proteins and biomaterials [67], mesenchymal stromal cells [68]/Implant devices, drug delivery, model surfaces. |
| **Non-Laser Techniques** | | | |
| Radio frequency magnetron sputtering (RF-MS) | Stoichiometric transfer, uniform dense coatings, good versatility of experimental design. | Limited to inorganic coatings, amorphous coatings, rather expensive. | Inorganic coatings: CaP [69,70], BG [71–73]/Implant devices. |
| Pulsed electron deposition (PED) | Stoichiometric and adherent thin films, low cost compared to other PVD techniques. | Limited to inorganic coatings, small area covering. | Inorganic coatings: HA, CaP, biogenic CaP, BG [74]/Implant devices. |
| Plasma spray (PS) | Simplest, operates at low costs. | Use for inorganic coatings only, poor coating adhesion, weak bonding strength at ceramic-metal interface. | Inorganic coatings: HA [75,76], and BG [77]/Implant devices. |

**Table 1.** *Cont.*

| Deposition Method | Advantages | Drawbacks | Bio-Coating Materials/Application |
|---|---|---|---|
| Layer-by-layer (LBL) | Good thickness control, viscoelasticity/bioactivity, coat multiple substrates of all scales. | Poor layers adhesion to substrates, difficult to create multilayers due to solvent issues, difficult to generate gradient coatings. | Organic coatings: biomaterials for drug delivery [78]/Drug delivery, model surfaces. |
| Langmuir-Blodgett dip coating | Possibility to assemble monolayers, high spatial coverage. | Limited to very thin films, typically used for organic coatings only. | Organic coatings: PCL-Blend-PEG [79]/Drug delivery, model surfaces. |
| Adsorption on surface | Simple, rapid and inexpensive, useful for organic bio-coatings. | Poor adhesion on substrates, poor uniformity, difficult to create multi-layer assembling. | Organic coatings: BMP-2, BMP-6, BMP-7 with fibronectin onto HA coatings [80]/Implant devices, drug delivery. |
| Spin coating | Simple and inexpensive, uniform coatings, accurate thickness control, high spatial coverage. | Use for 2D surfaces only, typically used for organic coatings, solvent issue in case of multilayers, poor adherence. | Organic coatings: protein-polysaccharide thin films [81]/Implant devices, drug delivery. |
| Sol-gel (SG) | Available for both organic and inorganic coatings, simple operation, high versatility. | Poor adhesion to substrates, difficult to create multilayers due to solvent issues, difficult to generate gradient coatings. | Inorganic, organic, hybrid coatings: HA, BG [82], Gentamicin/Chitosan/BG composite [83]/Implant devices, drug delivery. |
| Electrophoretic deposition (EPD) | Useful for both organic and inorganic coatings, simple processing setup. | Difficult to create multilayers due to solvent issues, poor coating adhesion. | Inorganic, organic, hybrid coatings: HA–iron oxide–chitosan composite [84,85]/Implant devices, drug delivery. |

### 3.2. Bio-Coating Adhesion Issues

Aside from specific characteristics of each technological process either applied to inorganic, organic, or composite materials, there is a strong correlation between the structure–property relationship and adhesion strength of synthesized coatings. Indeed, in the case of metallic implants for dentistry or orthopedics, a poor adherence of the covering thin layers could stand for premature delamination and eventually failure of the implant. For example, plasma-sprayed HA coatings contain stresses that induce a large mismatch between thermal expansion coefficients of the coating and metal substrate [86]. Typically, the adhesion strength of the bio-coatings is carried out by scratch, nanoindentation tests, and pull-off tests. Modifications of the substrate surface or the uses of an intermediate buffer layer play a significant role in the coating growth process and adhesion, particularly for inorganic materials. Specifically, sand-blasting, followed by acid-etching of a titanium surface, showed the influence of nano/micro-structure on grain size, mechanical properties, and surface wettability of Ag-doped HA bio-coatings deposited by the radio frequency magnetron sputtering (RF-MS) process [87]. In this study, nanoindentation tests revealed significantly higher nanohardness and Young's modulus values with decreasing grain size. The introduction of an intermediate buffer layer (TiN, $ZrO_2$, or $Al_2O_3$) between the Ti alloy substrate and HA coating deposited by PLD allowed for thin films with improved mechanical characteristics to be obtained compared with structures synthesized without the transitional layer [88]. In this case, the friction coefficient was found to be reduced by 25% when an intermediate layer was used. In addition, a direct comparison of RF-MS and PLD demonstrated that HA coatings exhibited improved mechanical properties when grown by both methods on Ti alloy substrates previously coated with a TiN buffer layer [89]. Generally, for HA coatings deposited by PLD, good adhesion properties are described when the substrates are heated during coating processes and samples are further thermally treated post-deposition [90]. However, pull-off tests revealed, in some cases, a decrease of tensile strength value with increasing substrate temperature from 480 to 550 °C during the deposition process [91]. A comprehensive comparative investigation on the adhesion of HA coatings obtained by different PVD techniques on Ti alloy substrates was reported by Mohseni et al. [92]. On the other hand, biopolymer adhesion to metallic substrates was found to be good in most cases [93]. Aside from material chemistry, surface energy was proven to be a dominant factor to induce in vitro cell adhesion

and proliferation, and found to play a rather more important role than surface roughness for cell colonization onto engineered tissue scaffolds [94].

## 4. Biomimetic Laser Processing

Laser-based technologies have been demonstrated to be a clean, fast, and cost-effective alternative to the existing procedures for organic, inorganic, and composite bio-coatings. They are in the forefront of several discoveries in nanoscience and nanotechnology. Laser-based deposition techniques have been widely employed for the fabrication of various thin biomimetic composite coatings [7,63,95–98]. The book edited by Schmidt and Belegratis [99] presents a detailed state-of-the-art of the laser technologies until seven years ago as well as the materials that are typically employed in cutting-edge biomimetic applications.

Two conventional laser approaches used nowadays for assembling biomimetic thin coatings are: (i) pulsed laser deposition (PLD) for the synthesis of inorganic coatings and (ii) matrix assisted pulsed laser evaporation (MAPLE) for the fabrication of organic, inorganic, and hybrid organic–inorganic coatings. Both techniques use pulsed laser beams to ablate solid targets in vacuum, deposit the ablated material cluster by cluster on facing substrates, and form a thin coating. By readapting setup configurations, in particular employing multiple pulsed laser beams and/or multi-targets, combinatorial-PLD and combinatorial-MAPLE were developed to fabricate thin blended coatings with a gradient of composition in a single-step process. The key benefits and drawbacks of these techniques have been previously addressed in a comparison with other non-laser-based methods [7,62,63,97] and are briefly summarized in the following. Both methods are highly versatile, with individual process parameters able to control and tailor morphological and structural coating properties. The use of laser beams for target irradiation in vacuum allows for material processing in a contamination-free environment, suitable for biomedical applications.

### 4.1. Pulsed Laser Deposition

In case of PLD, the main advantages refer to the possibility of preserving the material stoichiometry during transfer, but also to switching the composition chemistry and structure to generate new properties. During laser irradiation, a plasma plume is formed. This plume drives the atoms, ions, and nanoparticles from the ablated target to a facing collector where a thin uniform coating grows additively. The thickness is controlled by the number of applied laser pulses. The synthesis of new materials can be achieved by introducing different gases and controlling the pressure inside the chamber. This is a reactive process that was named reactive-PLD (R-PLD) due to the chemical reactions in the plasma phase. Pioneering work reported the synthesis of TiN thin films by laser ablation of a Ti target in an $N_2$ gas that filled the reaction chamber [100]. Highly adherent thin coatings can be obtained due to the high kinetic energies of ablated species when hitting a heated substrate. Multilayered coatings with predefined thickness and/or doped structures can be eventually fabricated by a two-step process in which distinct targets are ablated successively [101].

### 4.2. Matrix Assisted Pulsed Laser Evaporation

MAPLE process uses milder conditions (e.g., laser energies one order of magnitude lower than in case of PLD), just above the threshold of frozen target vaporization. It therefore allows for the fabrication of either inorganic, organic, or hybrid coatings with a high versatility [59,96,97,102–105]. MAPLE offers specific benefits with respect to PLD, demonstrated by the safe transfer and deposition of delicate compounds such as proteins [106,107] or biopolymers, without impeding on the stability of their functional characteristics, which rely on preserving their nanoscale structure [103].

The MAPLE technique was first introduced twenty years ago by McGill and Chrisey [108] at the U.S. Naval Research Laboratory in order to attain damage-free pulsed laser evaporation of organic materials and thin films in the late 1990s. It was designed as an alternative to PLD in order to avoid organic material decomposition, degradation, and/or denaturation induced by

high laser powers during ablation. Since then, MAPLE has evolved to produce a broad spectrum of organic, inorganic, and hybrid coatings [109–112] for various biomedical applications as well as for energy [113–115], sensing [116–118], wearable electronics, and photonic devices [98,119]. Indeed, although initially designed for polymers, the method proved successful for a large variety of compounds such as proteins [106,107], enzymes [120,121], polysaccharides [103,122,123], calcium phosphates [24,55,124,125], nanoparticles [126,127], proteins and drug functionalized graphene oxide [65], or carbon nanostructures embedded in organic matrices [128]. Its maturity has already been achieved and existing commercial installations fulfil the anticipations of the scientific community [129].

MAPLE is an additive physical vapor deposition process that was extensively explored for the functionalization of solid substrates with various coatings. In its most used configuration, the MAPLE procedure is based on pulsed laser irradiation of a cryogenic target using a UV laser beam (Figure 2). Typically, the target is composed of solute molecules that are dissolved in an appropriate solvent, compatible with the employed laser evaporation wavelength. During MAPLE processing, the frozen solute biomolecules are then transferred and assembled onto a receiving substrate, typically placed parallel with respect to the target at a specific separation distance (few centimeters in most cases). In the case of MAPLE, the evaporated biomolecules are collected on solid substrates, pulse by pulse, generating a thin coating with thicknesses from a few to several hundreds of nanometers. The targets are prepared by dissolving the active material in an appropriate solvent (e.g., water, chloroform, DMSO), followed by immersion of the solution in liquid nitrogen (LN) for an optimized time (typically 5–15 min, dependent on the solvent–solute pair type). After target solidification, this is placed inside a customized or commercial stainless steel reaction chamber. At this stage, the deposition parameters are set: target-to-substrates separation distance (3–5 cm), substrate temperature (room temperature or gentle heating), the nature and the pressure inside the reaction chamber (low vacuum *vs.* inert or reactive gas atmosphere). The most employed laser sources are excimer lasers, typically used for PLD. Setups and experiments in this review used excimer lasers (e.g., KrF*, $\lambda$ = 248 nm, with pulse duration $\tau_{FWHM}$ = 25 ns, operated at $\upsilon$ = 1–40 Hz), but other laser sources have also proven successful for obtaining functional organic coatings (Nd:YAG pulses at $\lambda$ = 266 nm [105,114,118], $\lambda$ = 355 nm [117], or for resonant absorption by Er:YAG laser at $\lambda$ = 2.94 $\mu$m [130,131]. A critical step before the deposition protocol, mainly for biomedical applications, involves careful substrate cleaning in successive ultrasonic baths of acetone, ethanol, and deionized water (at least 15 min each).

Aside from deposition parameter control (laser wavelength, pulse duration, laser energy, laser spot focusing, repetition rate, target-to-substrate separation distance, substrate temperature, ambient conditions, number of applied laser pulses) one may also design distinct substrate arrangements in various geometries. The literature suggests both on-axis and off-axis configurations [132], which are necessary to precisely control the thickness and spreading of the coating. Such geometries should be further correlated with substrate positioning with respect to target rotation (Figure 2).

Two setup configurations can be used: (i) samples with the same composition and thickness are obtained by rotating both the target and the substrates (Figure 2a), and (ii) samples with compositional gradient and distinct thicknesses are prepared by rotating the target and keeping the substrates fixed (Figure 2b). In the latter case, the main evaporation flux is perpendicular to the closest sample C1, while a concentration gradient toward C4 is naturally achieved due to material spreading along the radial–orthogonal direction of the substrates (x direction in Figure 2b). The goal of adopting such irradiation geometry is necessary when the generation of compositional gradient coatings is applied in a single-step process. One possible application revealed a possibility of investigating a drug dose-dependent effect when incorporated into the coatings, with respect to controlled delivery for cancer studies [65].

Indeed, it is generally accepted that when compared to other conventional, non-laser deposition methods (e.g., drop-casting, spin-coating, dip-coating, Langmuir–Blodgett), the MAPLE technique allows for high experimental versatility and somehow high control of coating thickness challenging also ultrathin film structures. Other important advantages refers to the possibility of preserving

structural and functional properties, even for very delicate compounds. The method was shown to easily provide congruent transfer of adherent and uniform coatings on centimeter sized substrates, micro-fabrication of multilayers from a multi-target system, and the possibility of fabricating gradient thin films in a single-step process. However, the main drawbacks are still related to difficulties met when large-area coatings (tens of centimeters) should be uniformly covered.

UV-MAPLE alternatives such as Resonant Infrared- (RIR-) and emulsion-based RIR-MAPLE including fundamental phenomena of laser beam interaction with frozen target and various applications have been previously reviewed elsewhere [98]. Briefly, the goal of using infrared (IR) wavelengths is to adjust the target absorption to a resonant region specific to the molecules of the solvent matrix, thus minimizing laser interaction with the materials to be transferred. Indeed, most of the vibrational frequencies of organic solvents are in the infrared region, and thus IR lasers can represent a viable alternative to UV lasers. Here, the IR excitation could be easily adapted for resonant interaction with specific chemical vibrational modes within the most used solvent matrix, and consequently, organic molecules experience minimal photochemical and structural degradation. A simplified schematic representation of the laser interaction processes applied to polymers using PLD, UV-MAPLE, RIR-MAPLE, and emulsion-based RIR-MAPLE techniques is presented in Figure 1.

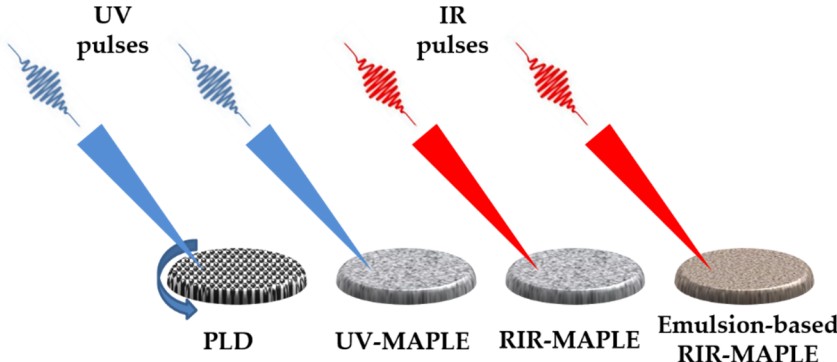

**Figure 1.** Simplified schematic sketch of the main laser-based deposition techniques of polymers: pulsed laser deposition (PLD), ultraviolet-matrix assisted pulsed laser evaporation (UV-MAPLE), resonant infrared-matrix assisted pulsed laser evaporation (RIR-MAPLE), and emulsion-based RIR-MAPLE techniques when compared to laser wavelengths (UV *vs.* IR) and target composition (solid pellet in case of PLD *vs.* cryogenic targets of frozen solutions/emulsions in case of MAPLE). Reprinted with permission from [98]. Copyright 2017 AIP.

The main differences between the processes are related to the physical–chemical processes at the laser–target interaction level. Polymer targets used with PLD typically consists of pressed powder or pellets, while MAPLE targets are composed of frozen solution. In the particular case of emulsion-based RIR-MAPLE, oil-in-water and/or water-in-oil emulsions are used in correlation with the hydrophobic/hydrophilic nature of the polymers to be grown as thin coatings. Few molecular dynamics simulation studies have addressed both UV- and RIR-MAPLE [133–135]. Another coarse-grained chemical reaction model [136] and a semi-empirical model based on the thermodynamics and kinetics of phase transitions in frozen solvent matrices [137] were evaluated for the complex transfer of organic molecules in MAPLE process.

## 5. Bio-Coatings with Multilayer Configurations and Gradient of Composition by Laser Deposition

In this section, we present a few representative examples of composite biomimetic coatings fabricated by laser techniques. Sima et al. [96] showed that multilayered inorganic–organic thin implant coatings could be obtained in a two-step laser process. PLD was first employed for growing HA thin films onto titanium substrates. Furthermore, fibronectin (FN) coatings were deposited by MAPLE

on top of HA layers. Using a cryogenic temperature-programmed desorption mass spectrometry analysis, the authors found that less than 7 µg FN per cm$^2$ HA surface is an optimum concentration for improving adhesion, spreading, and differentiation of osteoprogenitor cells. Moreover, the possibility of using gradient coatings to study drug delivery influence when interacting with melanoma cells was evidenced [65]. BSA-functionalized graphene oxide nanomaterials (GONB) incorporating inhibitory drugs demonstrated an efficient dose-dependent effect on melanoma cells.

### 5.1. Combinatorial Laser Technologies

Combinatorial MAPLE (C-MAPLE) was first introduced in 2012 by Sima et al. [138] in order to fabricate coatings with compositional gradients on large substrate areas (several cm), in a single-step process. The method could be considered an *extension* of combinatorial-PLD (C-PLD), initially introduced by Takeuchi et al. [139], for the fabrication of inorganic compositional library thin films. Few C-PLD experiments aimed to generate inorganic bioactive coatings with gradient of composition for tissue engineering have been reported [57,58]. Socol et al. [57] have shown the possibility of controlling the composition and surface morphology characteristics in the case of Ag-doped CaP thin films by using C-PLD. Later, the same group reported the synthesis of antimicrobial libraries of Ag-doped carbon thin films, in order to be proposed as coatings that could minimize the risk of implant-associated infections [58].

C-MAPLE proved an appropriate method for the synthesis of several bio-coatings with compositional gradients and morphology for either organic, inorganic, or hybrid nanostructures with the view of creating biomimetic microenvironments. Representatives examples refer to protein embedded in biodegradable polymers [43], polysaccharides [64,123], ion and drug doped CaPs [25,61,140], or graphene oxide for controlled drug release [65].

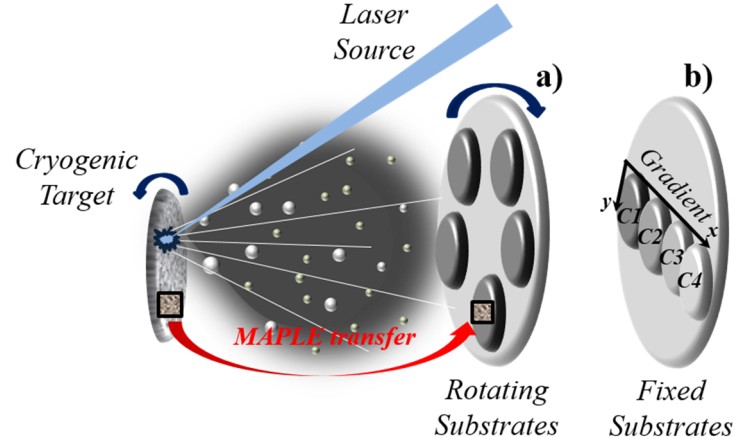

**Figure 2.** Schematic representation of the main deposition geometries in the case of MAPLE used to obtain coatings with uniform composition (**a**) or coatings with gradients of composition (**b**). Adapted from [65].

The experimental design employed in the case of C-MAPLE as well as the main advantages and limitations are explained in the following. In a typical irradiation geometry, two cryogenic targets are simultaneously evaporated by two pulsed laser beams. In our experimental setup, one laser beam was split and focused onto the targets. Alternatively, two distinct laser sources with different emission characteristics (wavelength, pulse duration, repetition rate) could be employed. A schematic representation of experimental setup is presented in Figure 3a. The plumes containing evaporated materials are subsequently collected on the receiving substrates, similar to MAPLE deposition. Due to plumes spreading and mixing, a gradient of composition is achieved along the longitudinal direction of the substrates, thus generating a combinatorial library as schematically depicted in Figure 3b. Fast and controlled gradients could be obtained on large areas (e.g., a glass slide scale) by adjusting the target to

the substrate distance and the separation distance in-between the irradiation spots (*S* and *D* parameters in Figure 3b).

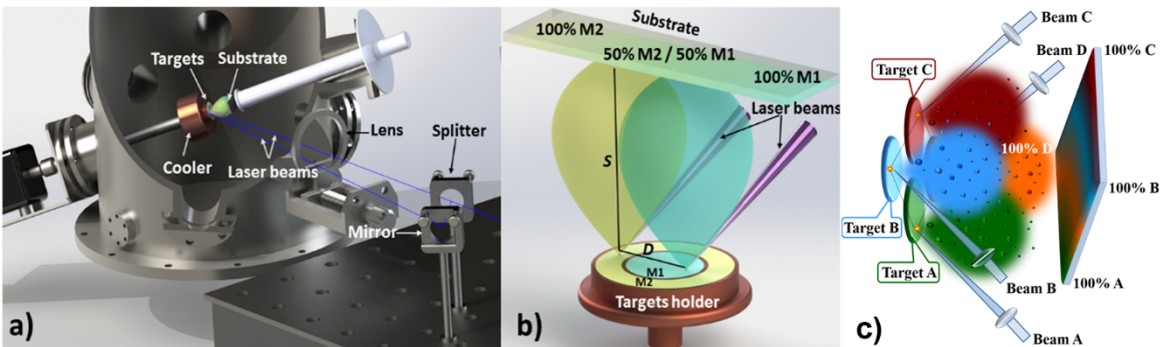

**Figure 3.** Experimental design of the C-MAPLE thin films processing (**a**); detailed representation of the target–substrate arrangements for combinatorial materials distribution (**b**); advanced experimental design for tailoring combinatorial diagram maps on large 2D substrates (**c**). Courtesy of M. Sopronyi (**a** and **b**) and adapted (**c**) from [63].

An advanced experimental design could be envisioned in order to generate multi-compositional diagram 2D maps on high surface areas. In this special configuration, several targets should be simultaneously irradiated using predefined laser parameters (Figure 3c). Consequently, by precisely tuning the experimental conditions, one could fabricate complex combinatorial coatings in a single-step process for high-throughput screening of the composition–structure–properties relationship at every point of the surface. Moreover, new composite materials and improved functionalities could be foreseen (e.g., drug mixtures for new therapeutic strategies).

Indeed, the basic concept of combinatorial materials science relies on synergistic mechanisms in which the interplay of two compounds with distinct properties create a novel functionality. The literature proposes various approaches to fabricate combinatorial coatings, most of them being suitable only to a limited class of compounds. Among these, we can mention several physical–chemical deposition methods such as magnetron co-sputtering deposition [141], glancing angle deposition [142], co-electrodeposition [143], casting processes [144], or flow-coating methods [145].

*5.2. Ion-Doped Inorganic Bio-Coatings Obtained by Combinatorial-Matrix Assisted Pulsed Laser Evaporation*

CaPs containing various amounts of different ions incorporated in the apatite crystal structure were proposed in recent research [146,147]. Among several useful divalent ions, Sr has evidenced positive effects on bone metabolism, and the introduction of Sr ranelate and use for potential treatment of osteoporosis has been demonstrated [148,149]. Zinc (Zn) also plays an essential beneficial role on reducing the risk of osteoporosis by bio-mineralization [150]. Among CaPs, β-tricalcium phosphate (β-TCP) exhibits greater solubility and resorbability compared with HA [151]. Chou et al. showed that biomimetic Zn-βTCP can stimulate faster osteogenic differentiation of MSCs to osteoblasts than pure β-TCP [152]. More recently, Boanini et al. [140] demonstrated that the C-MAPLE technique could be an alternative approach for the synthesis of gradient coatings of Sr doped HA–Zn doped β-TCP. The irradiation scheme is depicted in Figure 4a. The depositions were performed on Ti discs with a 12 mm diameter, labeled from A to E. The compositional gradient was then generated in-between A, which corresponded to 100% Sr:HA and E, which corresponded to 100% Zn:β-TCP (Figure 4a). This allowed for the demonstration of the concrete influence of the composition, structure, and topography of the implant surface on the osseointegration using an in vitro co-culture model. The preservation of the crystalline phases of the bio-coatings (with respect to the as-prepared powders used for targets) with distinct morphological features and the compositional distribution of the dopants along the gradient coatings was validated. As shown in Figure 4b, samples A and E contained Sr (green) and Zn (blue) only, while a homogenous mixture of them was naturally achieved in-between.

Human 2T-110 osteoclast precursors and human MG-63 osteoblast-like cells were co-cultured on gradient samples (arrows on Scanning Electron Microscopy - SEM images presented in Figure 4c). The response of cells was modulated by the gradual composition and strongly influenced by the Sr:HA/Zn:β-TCP ratio. In particular, Sr:HA was found to inhibit osteoclast viability and differentiation, while Zn:β-TCP proved a beneficial role on the mineralization process. The central regions of the bio-coatings exhibited rather combined effects on either osteoblast or osteoclast cells. It was concluded that gradient bio-coatings could provide materials with improved functionalities, in order to enhance and accelerate bone repair.

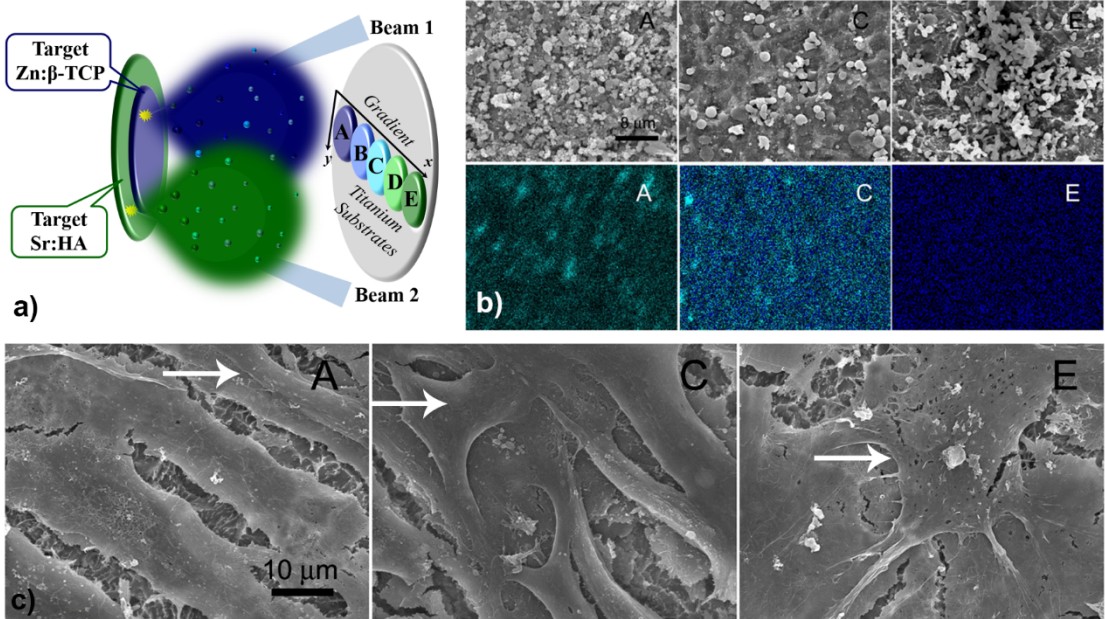

**Figure 4.** Experimental design for Sr:HA to Zn:β-TCP compositional gradient generation (**a**); scanning electron microscopy (SEM) images and Energy Dispersive Spectroscopy (EDS) maps of A, C and E thin films as-prepared within the compositional gradient. In the maps, the color codes represent Sr in green and Zn in blue (**b**); SEM images of osteoblasts grown on A, C and E thin films, respectively, at seven days post-seeding (**c**). Arrows in (c) indicate the bone cells. Reprinted with permission from [63,140]. (Copyright 2018 Elsevier).

### 5.3. Multi-Functional Organic Bio-Coatings Obtained by C-MAPLE

C-MAPLE was recently proposed for the synthesis of gradient biopolymer thin film assemblies [64] for tissue engineering and regenerative medicine applications. Synchronized laser evaporation of cryogenic targets containing sulfated *Halomonas* levan (SHL) and quaternized low molecular weight chitosan (QCH) revealed successful fabrication of combinatorial gradient bio-coatings. Studies have shown that levan polysaccharides are highly efficient in several biomedical applications such as antioxidant and anticancer activities, drug carrier systems, bioactive thin film blends, multilayer adhesive films, but also temperature responsive and cytocompatible hydrogels [153–156]. Recently, SHL has shown heparin mimetic anticoagulant activity [157] as well as improved mechanical and adhesive properties of cytocompatible and myoconductive films for cardiac tissue engineering applications [158]. Chitosan is a natural biomaterial, applied in a wide range of biomedical applications such as wound healing or tissue engineering [159,160], implant coatings [161,162], and drug delivery systems [151,163].

Mihailescu et al. [64] have demonstrated that combinatorial libraries of SHL/QCH could influence mouse fibroblasts viability (L929 cell line), coagulation effects with respect to SHL contents, and antimicrobial activity of the gradients against *Escherichia coli* and *Staphylococcus aureus* strains. The C-MAPLE design is presented in Figure 5a. Four silicon or glass substrates were positioned in front of the two evaporated plumes to be coated with composite SHL/QCH). SEM analyses of samples

revealed morphological differences along the combinatorial library (Figure 5b). Specific features to each compound were observed. The irregular, interconnected polymer networks of quasi-spherical particles found in the case of SHL regions gradually disappeared with an increase in QCH content. The cell proliferation assays demonstrated good biocompatibility for all samples, with a predominance for samples consisting of 75% QCH/25% SHL (Figure 5c). This study also evidenced the highest blood clotting speed on SHL regions, which decreased with QCH content. Fluorescence microscopy images (Figure 5d) showed normal cell morphology, with polyhedral and elongated shapes, which is in agreement with the proliferation assays. The anti-biofilm activity assays revealed inhomogeneous bacterial cell adhesion at both early (8 h) and late (24 h) time points for *E. coli* along the SHL–QCH reciprocal gradient (Figure 5f). However, *S. aureus* strain tests showed that the biofilm was rather inhibited by the samples containing the highest amounts of either levan or chitosan (Figure 5e).

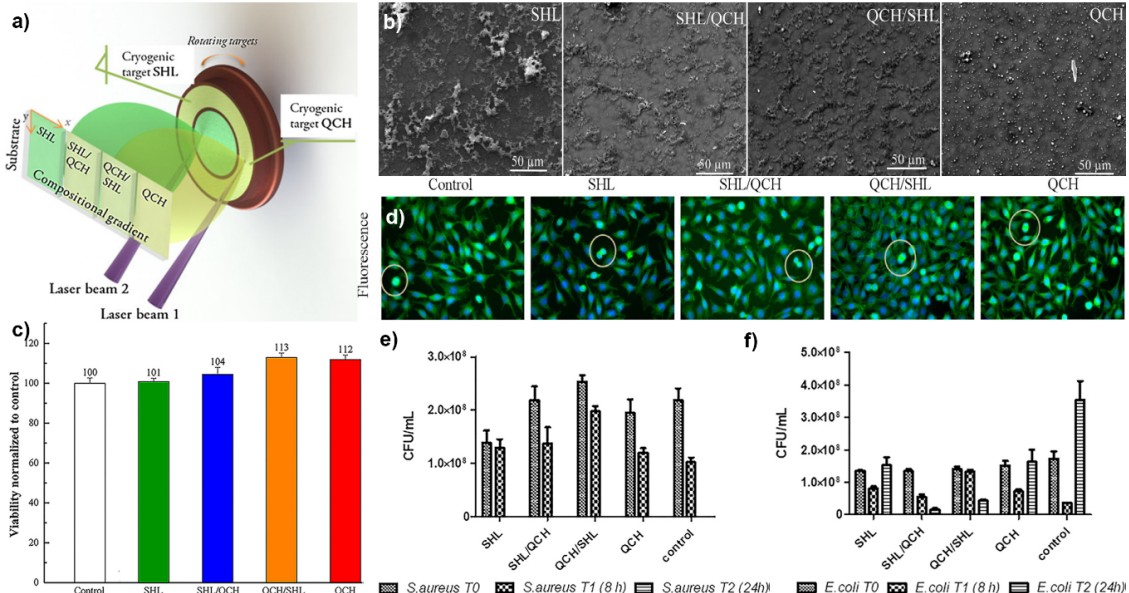

**Figure 5.** Experimental C-MAPLE setup (**a**); SEM micrographs of sulfated *Halomonas* levan (SHL) (**left**) to quaternized low molecular weight chitosan (QCH) (**right**) gradient bio-coatings (**b**); L929 cell proliferation on combinatorial coatings compared to the glass control (**c**). Fluorescence microscopy images of L929 cells onto the control (glass) and gradient coatings (magnification ×40). Actin fibers in green are marked with Fluorescein-Phalloidin, while nuclei in blue are stained with Hoechst. (**d**); Temporal dynamics of the *S. aureus* (**e**) and *E. coli* (**f**) biofilms growth on the C-MAPLE coatings. Circles in the fluorescence images (**d**) represent cells in various stages of division. Reprinted with permission from [64]. (Copyright 2019 Elsevier).

## 5.4. Hybrid Bio-Coatings by MAPLE for Anti-Tumor Drug Delivery to Cancer Cells

Very recently, Sima et al. [65] reported on the successful laser synthesis of graphene oxide nanomaterials (GON) and hybrid GON-BSA (GONB) nano-coatings incorporating anti-tumor drugs targeting melanoma cells. Among the deadliest forms of cancer, melanoma is the most aggressive skin cancer due to its high multi-drug resistance, frequent relapse, and decreased survival rates [164]. Due to their unique structural, physical, and chemical properties, carbon-based nanomaterials are extensively explored for drug/gene delivery in cancer therapy, bacteria-killing, tissue engineering platforms, engineering stem cell responses, biosensing, and cellular imaging [8]. On the other hand, nanomaterials, both synthetic and natural, may have side-effects on human tissues and generally for health, while the exposure risks are difficult to precisely evaluate [165].

In our experiments, Dabrafenib (DAB) and Trichostatin A (TSA) drugs, inhibitors for melanoma cell molecular targets, were introduced within the thin GONB coatings with successful preservation of their functional properties. DAB (chemical formula $C_{23}H_{20}F_3N_5O_2S_2$) is a drug approved by the Food

and Drug Administration (FDA) in 2013, for the treatment of advanced metastatic melanoma bearing the mutated BRAF gene (BRAFV600E) found in about 70% of melanoma tumors [166]. This inhibitor proved efficient in clinical trials of phase 1 and 2 in patients with BRAFV600E mutated metastatic melanoma [167]. TSA (chemical formula $C_{17}H_{22}N_2O_3$) is a promising Histone Deacetylase (HDAC) epigenetic inhibitor used in first phase clinical trials for relapsed or refractory hematologic malignancies. HDAC inhibitors have been proven to enhance the efficacy of BRAF/MEK inhibitors in both sensitive and insensitive RAS pathway–driven melanomas [168]. A comprehensive in vitro cytotoxicity assay was conducted. For comparison, GONB solutions and coatings were tested against seven different cell lines (Table 2). It was found that for concentrations up to 37 μg/mL in water, the nanomaterials did not induce any significant decrease in cell viability for either normal or transformed cells. When exposed to the coatings, optimal cell viability was achieved on films obtained from targets that contained up to 12 μg/mL for GON and 111 μg/mL for GONB.

**Table 2.** Cell line characteristics.

| Cells | Malignancy Phenotype | Mutation Status | Pigmentation |
|-------|---------------------|-----------------|--------------|
| MNT-1 | Primary melanoma | BRAF V600E | Highly pigmented |
| SKmel28 | Primary melanoma | BRAF V600E | Amelanotic |
| MelJuSo | Primary melanoma | BRAF wt; N-Ras Q61K | Amelanotic |
| A375 | Metastatic melanoma | BRAF V600E | Amelanotic |
| SKmel23 | Metastatic melanoma | BRAF wt | Pigmented |
| NHEM | Normal primary melanocytes | - | Pigmented |
| HDF | Normal dermal fibroblasts | - | - |

The determined safe concentration windows were further considered as starting points for MAPLE target preparation and deposition of the coatings containing inhibitors. The experimental design is presented in Figure 6a. A compositional gradient was generated between samples C1 and C3 in Figure 6a. SEM analysis evidenced smooth surfaces in the case of GON thin films, while the presence of BSA was found to drastically alter the morphology of the coatings, which also resulted in higher surface roughness (Figure 6b). The successful deposition and functionalization of each GONB-drug hybrid coating was further demonstrated by evaluating: (i) cellular BRAF activity inhibition and (ii) histone deacetylases activity blocking. DAB inhibition activity was validated by the decreased ERK phosphorylation in the SKmel28 primary melanoma cell line, while the TSA effect was monitored by acetylated histone accumulation in SKmel23 metastatic melanoma cell nuclei. Hence, a dose-dependent effect on target activity was evidenced for melanoma cells exposed to GONB coatings with a compositional gradient of inhibitors (Figure 6c).

Indeed, the coating efficiency was proven twice: (i) It was shown that by increasing the concentrations of GONB-TSA, an increase in fluorescence signal intensity was observed and correlated with the proportion of SKmel23 cells expressing acetyl histone H3 in the nucleus (Figure 6c left panel) and (ii) it was evidenced that by increasing the concentrations of GONB-DAB, a proportional decrease in pERK signaling in SKmel28 cells was noticed (Figure 6c, right panel) when compared to cells exposed to GONB coatings only (Figure 6c, top images). At the same time, immunofluorescence studies clearly evidenced a reduction in cell density on GONB bio-coatings loaded with a higher concentration of inhibitors (corresponding to the C1 samples).

These promising results stimulate further studies to strive for a deeper understanding of the mechanisms of laser-based coatings targeting melanoma and other cancers, with direct applications in personalized therapy studies. Such combinatorial bio-platforms could present high potential for screening cell-biomaterial interface activity for a broad spectrum of biomedical applications.

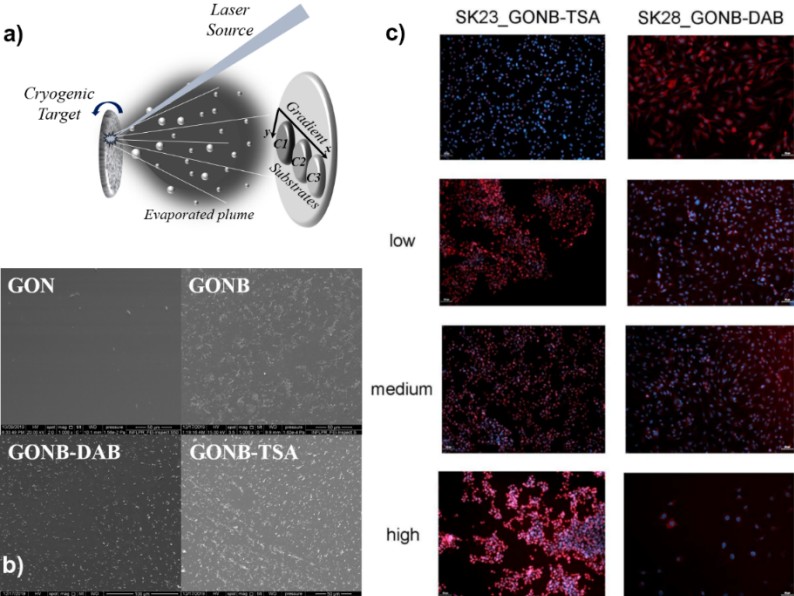

**Figure 6.** Experimental MAPLE setup for the synthesis of GONB-DAB/GONB-TSA composite bio-coatings with compositional gradient (fixed substrates) (**a**); SEM images of thin films deposited on silicon substrates (**b**); GONB-DAB and GONB-TSA gradients (from C3 low to C1 high) tested by immunofluorescence microscopy in comparison with simple GONB coated substrates (**c**). Here, the expressions of acetylated histone H3 (left panel) and pERK (right panel), respectively, are labeled in red, while the nuclei are labeled in blue (Hoechst). Scale bar = 50 μm. Adapted from [65].

## 6. Conclusions

Various technologies and methods have been applied to obtain thin bio-coatings at the interface between inert substrates and close cellular microenvironments. Most of the processes have been developed with the aim to produce either single layer, composite, or multi-layer coatings, with application specificity. Technological advancements have encouraged research to challenge biomimetic environments. Laser deposition techniques have been successfully used to fabricate bio-coatings, in particular pulsed laser deposition for inorganic materials and matrix assisted pulsed laser evaporation for both organic and inorganic materials. They have since progressed and were found applicable to either inorganic, organic, or inorganic–organic multi-layers or blended bio-coatings. Thus, combinatorial-PLD and combinatorial-MAPLE were proposed as alternatives to classic combinatorial chemical and physical deposition methods to synthesize biomimetic assemblies of complex composite and hybrid materials, with gradient of composition. C-MAPLE offers the unique characteristic of combining, in a controlled process, blended or multi-layer coating configurations of compounds dissolved in different solvent solutions, without the impediment of not being able to choose combinations with non-mixable solvents. Such combinations could allow for the synthesis of new materials with properties close to those of the native biological environment. in vitro evaluations of the bio-coatings fabricated by laser technologies confirmed their biocompatibility and capacity of modulating cell behavior. By combining the newly developed laser technologies with other chemical or physical methods, great perspectives could be open for domains like tissue engineering, nanomedicine, or controlled drug delivery in cancer research.

**Funding:** Partial financial support was given by the Romanian Ministry of Education and Research, under Romanian National Nucleu Program LAPLAS VI (contract No. 16N/2019) and by the Structural and Functional Proteomics Research Program of the Institute of Biochemistry of the Romanian Academy.

**Acknowledgments:** This review was entirely conceived, written, and processed under lockdown conditions during the COVID-19 global pandemic.We acknowledge UEFISCDI, Grant Nos: PCCDI63/2018 (PN-III-P1-1.2-PCCDI2017-0728) and TE7/2018.

**Conflicts of Interest:** The authors declare no conflicts of interest.

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
