# Peer review of "Biomimetic Coatings Obtained by Combinatorial Laser Technologies"

_coatings, doi:10.3390/coatings10050463_

Round 1

Reviewer 1 Report

I would recommend this review paper for publication after editing of English language and style.

Author Response

The authors did their best and have extensively reviewed the English language of the manuscript. The modifications are highlighted.

Reviewer 2 Report

Dear Authors,

The article shows comprehensive review of the multilayered biocoatings grown under laser control. The work is well written, however please correct the following:

- page 5, line 144, there is:  ossointegration, it should be osseointegration

- please write Figure 6 caption under the figure – in the present form the figure caption is inside body text on page 14, lines 538-543

Author Response

We have corrected the word ossointegration. After manuscript modifications, the correction appears now at line 140 in the highlighted version of the manuscript (and line 111 version without highlights).

Figure 6 caption is now placed at right place. 

Reviewer 3 Report

Emanuel et.al review laser-based combinatorial synthesis of bio-coatings for tissue engineering and cancer research. However, the article has some serious flaws as the manuscript discusses other bio-coatings techniques i.e inkjet printing, spin coating, and other strategies that do not match the title. Moreover, I have several comments about the same to improve the manuscript.

1) Abstract: The authors should follow the journal guideline (i.e the abstract should be a total of about 200 words maximum). The abstract should contain the main objectives and results of the review article.

2) Introduction:  I would strongly suggest that the author modify the introduction in line with the title

a) The authors should focus on Laser Technologies and its application in bio coating for tissue engineering.

b) Table 1 should have studies only focused on Laser Technologies

c) Biomimetic strategies for tissue engineering and cancer research should be discussed briefly in the introduction before introducing the advantages of laser our other techniques. I would strongly suggest that the author should focus only on tissue engineering and regenerative strategies

3) The other sections need to be rearranged to give the reader the advancement in laser technology and the process to grow thin bio-coatings for tissue regeneration strategies.

Author Response

We are grateful to the reviewer for he careful reading and useful observations that helped us to improve the manuscript.

In response to the points raised by the reviewer:

  1. We have reduced the Abstract below 200 words and underlined the main objectives.
  2. We have slightly changed the title and the manuscript organization. We hope that the new title and organization of our review paper will respond to the reviewer requests and also to the aim of the Special Issue.
  3. With this revised version, we have tried to focus on Laser Technologies within the manuscript. We also have planned to compare the laser technologies with some other Plasma Vapor Deposition (PVD) methods as it is in the aim of the Special Issue as well as methods applied for organic coatings. For example, in the “In vitro testing strategies” Section there are comparisons between plasma spraying technique (known as standard for coating with hydroxyapatite coatings), with other PVD technologies with emphasize on laser depositions. The re-organization of the manuscript could help readers to follow easier the text.
  4. We have re-organized Table 1 and focused on Laser Technologies, as requested. Nevertheless, as mentioned above, we have kindly ask the reviewer to accept few other techniques for comparison only. We have also deleted two methods from the Table.
  5. After re-organization of the text, our paper is focused on tissue engineering and regenerative strategies, while the applications of bio-coatings for cancer research covers a minor part only. We would like to keep this part in order to underline the versatility of laser technologies to obtain bio-coatings for another use and with eventual potential for targeting combined applications of tissue engineering and cancer research.
  6. All Sections were re-arranged and sub-divided. We hope the readers will benefit both of understanding the advancement in laser technology and of bio-coatings for biomedical applications.

Reviewer 4 Report

This review article introduced the advancement of laser technologies and their applications in biocoatings. The manuscript explains various laser-based approaches including pulsed laser deposition and matrix-assisted pulsed laser evaporation. The fundamentals of these approaches are well described in pages 7-9. The applications of biomimetic coatings by mentioned laser technologies are described and nicely organized in section 4. It contains a lot of information but at times it lacks details especially when examples are given.

Overall structure of the manuscript is not well organized, making it difficult to read and follow. Each section can be subdivided like section 4. Here are comments and suggestions:

-The title doesn’t quite represent the content of the manuscript. It shows not a clear indication of biomedical applications of laser technologies.

-The abstract is too long and contains too much information which is more appropriate in the introduction section.

-It seems that section 3, the part explaining PLD and MAPLE, should come after the introduction. There is a huge gap in understanding between introduction and section 2. So it may flow better to explain the laser techniques after the classical deposition techniques.

-Table 1 includes vast information, will it be possible to draw distinctions; inorganic, organic, and hybrid biocompatible coatings within the table? Also borders will make it easy to read. It will be helpful to include how these coatings are being used, e.g., which application? For implants or as drug delivery systems?

-Section 1 heading is missing on page 2.

-Section 2 needs to be subdivided, and paragraphs/contents describing biomimetic strategies should somehow relate to the coating process. It is difficult to make connections by reading this section. Also it needs to specify that it is bone tissue engineering that authors focused on.

-Line 66, it mentions a broad variety of coating materials. Could you explain whether it's coating of implants or else?

-Line 102, define BMP-2 signalling

-Lines 134-135, COLL1 and ALP, please explain what they are.

-Lines 156-157, were these epigenetic modulators coated?

-Line 203, is the equation necessary? Then it should be presented and explained better.

-Figure 2 legend should include a brief description of target materials even though it is explained in lines 349-353.

-Figure 4c and lines 441-444, it will be helpful to include arrows in the image indicating osteoblasts and details of the “combined effects”.

-Figure 5d, what are the circles indicating?

-Line 514, “GON”, please define GON.

-Figure 6 seemed to be misplaced as the legend for fig 6 appears lines 538-543.

-In conclusions- lines 567-569, authors should mention that studies presented in the manuscript are in vitro evaluations before drawing conclusions. 

Author Response

We are grateful to the reviewer for he careful reading and useful observations that helped us to improve the manuscript.

In response to the points raised by the reviewer:

  1. We have changed the title and the manuscript organization, as requested. All Sections were rearranged and sub-divided. We hope that the new title and organization of the review manuscript will respond to the reviewer requests and also to the aim of the Special Issue.
  2. We have reduced the Abstract below 200 words and underlined the main objectives.
  3. We temped another organization of the manuscript, starting with biomimetism and issues met in vitro and in vivo testing of bio-coatings. We hope the flow is easier to follow now and the readers will benefit on both of understanding the advancement in laser technology and of bio-coatings for biomedical applications.
  4. We re-arranged the Table and have focused it on Laser Technologies and specificities of applications to inorganic-organic-composite coatings.
  5. We have corrected most of the headings.
  6. We have subdivided the Section 2 and added a point of Clinical Trials.
  7. We have introduced “tissue engineering applications” for the coating materials, now at lines 71-72 in revised version with highlights (lines 49-50 in revised version without highlights).
  8. We have defined the BMP-2 abbreviation and shortly explained its role in bone physiology (lines 100-101 in revised version with highlights) – (lines 74-75 in revised version without highlights).
  9. We have defined and explained COLL1 and ALP markers role in MSCs differentiation (lines 128-130 in revised version with highlights) – (lines 100-101 in revised version without highlights).
  10. The epigenetic modulators were added to the cultures cells during a biochemical study of their potential beneficial role in stimulating MSCs osteogenic fate. For clarity, we have changed “pre-treatment” into “biochemical pre-treatment” (line 172 in revised version with highlights) (line 143 in revised version without highlights). This represents a proof-of-concept that MSCs can undergo differentiation pathway commitment change upon receiving external stimuli, which could help directing them towards therapeutically relevant fates.
  11. We removed the equation and let the piezoceramic material only in the text.
  12. We have introduced in Fig. 2 legend a brief description of laser wavelengths and of target materials, respectively.
  13. We added arrows indication osteoblasts on Fig. 4c. There is no evidence of combined effects on the Figures so that we have removed the reference in the text to the Figure.
  14. We added information in the Fig. 5 caption about the circles.
  15. We defined GON at lines 623-624 in revised version with highlights (lines 542-543 in revised version without highlights).
  16. We have corrected the place of Fig. 6 legend.
  17. We have introduced a new phrase in the Conclusions, lines 706-707 in revised version with highlights (lines 616-618 in revised version without highlights).

The authors did their best and have extensively reviewed the English language of the manuscript. The modifications are highlighted.

Reviewer 5 Report

  1. The authors are advised to provide some more details on the prospective use of HA-based coatings regarding their clinical relevance DOI: 10.1016/j.cobme.2019.02.003, ·  DOI: 10.1016/j.actbio.2013.10.036
  2. RF-magnetron sputtering was started in Russia by the group of prof. Pichugin, thus the reference 11 should be changed into DOI: 10.1016/j.surfcoat.2008.01.038
  3. I would request to improve the resolution of the figures. Some information is barely seen, e.g. Fig 5a, Fig3c.
  4. Could the authors provide some results of in vivo trials to reveal effectiveness of the approaches the review is devoted to?
  5. Some more details on structure-property-relationship regarding the adhesion strength should be provided.

Author Response

We are grateful to the reviewer for he careful reading and useful observations that helped us to improve the manuscript.

In response to the points raised by the reviewer:

  1. We have slightly changed the title and the manuscript organization, as requested. All Sections were rearranged and sub-divided. We also introduced a new paragraph on clinical relevance, new Sub-section 2.4, lines 248-278 in version with highlights (lines 199 – 229 in the revision version without highlights).
  2. We have introduced the new suggested reference in the Table (new ref. 69) after Table re-organization. It was not our intention to not mention the reference but only we intended to provide new references.
  3. All figures were modified to improve the resolution.
  4. We have introduced, as mentioned to the first point, a new paragraph on in vivo clinical trials (new Sub-section 2.4, lines 248-278 in revised version with highlights) - (lines 199 – 229 in the revision version without highlights).
  5. We have introduced a new paragraph, (new Section 3.2, lines 301-330 in revised version with highlights) – (lines 250 – 278 in revised version without highlights) about few details on structure-property-relationship regarding the adhesion strength.

Round 2

Reviewer 3 Report

Accept

Reviewer 4 Report

Authors did a good job addressing comments.

Reviewer 5 Report

can be accepted